# MMRC: Measuring Massive-computational math Reasoning with Code in LLMs

## Abstract

We introduce MMRC, a benchmark of 500 curated problems designed to evaluate large language models (LLMs) on their ability to integrate mathematical reasoning with code execution. Unlike existing benchmarks that emphasize either elementary or olympiad-level mathematics, MMRC focuses on university-level subjects including calculus, discrete mathematics, linear algebra, linear programming, mathematical physics, numerical computation, and scientific computing. Each problem is deliberately adapted to impose substantial computational workloads, making text-only reasoning infeasible and requiring code execution for accurate solutions. We evaluate 120 model configurations, spanning open- and closed-source models, on MMRC under two paradigms: Code-Invoked Reasoning (CIR), where models generate Python scripts, and Math-Code Agent (MCA), where models dynamically interact with a Python interpreter. Code integration consistently improves accuracy on complex tasks while reducing token usage, establishing MMRC as the first systematic benchmark for math–code integrated reasoning and advancing LLMs toward real-world problem solving.

## 1 Introduction

Recent advancements in large language models (LLMs) have transitioned them from basic language understanding to sophisticated logical reasoning (Wei et al., 2022b; Chowdhery et al., 2022), achieving significant success in STEM-related tasks (Comanici et al., 2025). Mathematics, as both a fundamental science and a rigorous benchmark, plays a pivotal role in evaluating the reasoning capabilities of these models (Hendrycks et al., 2021). However, as mathematical tasks become more complex, particularly in engineering applications, solutions often require intensive computation or large-scale simulations (Lewkowycz et al., 2022; Li et al., 2022). Text-based reasoning alone is insufficient for these tasks. Integrating textual reasoning with code execution allows models to maintain logical coherence while delegating computational tasks to code, resulting in more efficient and accurate outcomes compared to purely autoregressive methods(Gao et al., 2023; Schick et al., 2023; Yao et al., 2023).

Current benchmarks, as showed in Table 1, which cover a range of problems from elementary arithmetic to olympiad-level challenges, primarily evaluate text-based reasoning and symbolic manipulation. While these benchmarks are important for assessing basic reasoning skills, they fail to capture the full range of abilities required in real-world applications, where mathematical reasoning often involves the integration of algorithmic design and code execution. This gap highlights the need for a benchmark that evaluates a model's ability to combine reasoning with computational tasks, a capability that remains largely unaddressed in existing frameworks.

In this paper, we introduce MMRC, a benchmark of 500 problems designed to evaluate a model's ability to integrate textual reasoning with code execution. Unlike existing benchmarks that focus on advanced mathematical domains, MMRC emphasizes generality and progressive capability improvement. It covers university-level mathematics, including calculus, discrete mathematics, linear algebra, linear programming, mathematical physics, numerical computation, and scientific computing.

Each problem is crafted to impose substantial computational workloads, making purely textual reasoning infeasible. A representative sample is shown in Figure 1. As a result, models must invoke code execution to solve the problems accurately. In addition to increasing numerical complexity,

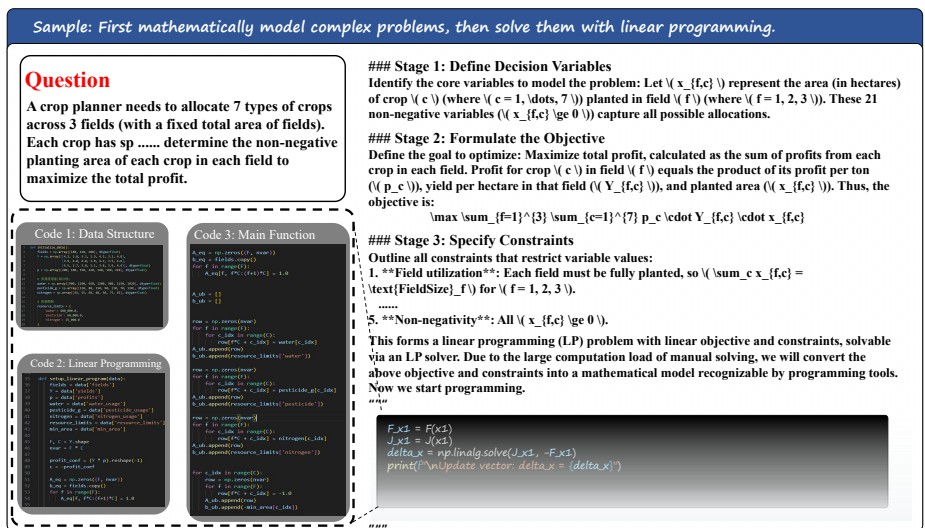

Figure 1: **A sample from the MMRC benchmark**. In this case, textual reasoning is indispensable for determining the exact quantity to be solved. However, once the target equation is identified, our tailored adaptation renders it extremely difficult, even impossible, for the model to solve purely in natural language. Thus, we guide the model to employ code for formal modeling and numerical computation, as illustrated by the use of dynamic programming in this example.

the problems introduce algorithmic challenges, such as recursive calls and numerical simulations. Authored by doctoral-level and master-level mathematics students, the problems were selected from an initial pool of 1,200, with each requiring at least three hours of adaptation, code integration, and cross-verification.

We evaluate 120 distinct model configurations, spanning open-source, closed-source, and math/code-specialized models, using the MMRC benchmark across two reasoning and code paradigms. In CIR mode, models generate and execute a Python script with reasoning embedded as comments. In MCA mode, the model interacts with a Python interpreter, dynamically deciding when to invoke code. Our analysis shows that models achieve substantially higher accuracy on the base set than on the hard set, confirming that the partition effectively captures varying levels of computational reliance. At the same time, code execution serves as a double-edged sword. On analytically tractable base problems, forcing models to invoke code can disrupt otherwise viable reasoning trajectories and could lead to accuracy drops. By contrast, on code-dependent hard problems, code becomes indispensable, consistently improving overall accuracy. In addition, on the hard set, code execution often reduces token costs by compressing lengthy textual derivations into concise scripts. Finally, error analysis reveals complementary weaknesses across inference modes: CIR highlights the brittleness of one-shot program generation due to syntax and implementation errors, whereas MCA suppresses low-level coding mistakes but exposes deeper flaws in abstract reasoning and problem understanding.

Overall, this work introduces the MMRC benchmark as a novel tool for evaluating LLMs, offering a more comprehensive evaluation that aligns closely with real-world problem-solving tasks.

## 2 RELATED WORK

**Mathematics Benchmarks.** Early math benchmarks like AddSub (Hosseini et al., 2014) and SingleEq (Koncel-Kedziorski et al., 2015) focused on basic arithmetic and algebra. Later, datasets such as GSM8K (Cobbe et al., 2021), SVAMP (Patel et al., 2021), and Multi-Arith (Roy & Roth, 2016) increased difficulty, followed by more challenging benchmarks like MATH (Hendrycks et al., 2021) and MMLU-Pro (Wang et al., 2024). Recently, even harder benchmarks like AIME (aim, 2024 URL https://maa.org/math-competitions/american-invitational-mathematics-examination-aime), Omni-Math (Gao et al., 2024), and FrontierMath (Glazer et al.,

Table 1: **Comparison of MMRC with existing mathematical reasoning benchmarks.** MMRC is designed at the university level with expert-authored, computation-intensive, and code-integrated problems.

| Benchmark | Design Characteristics | | | Level | Size | Eval Metrics |
| | Expert-Original | Comp.-Intensive | Code-Integrated | | | |
| --- | --- | --- | --- | --- | --- | --- |
| GSM8K (Cobbe et al., 2021) | ✗ | ✗ | ✗ | Grade School | 8.5k | Acc |
| MATH (Hendrycks et al., 2021) | ✗ | ✗ | ✗ | High School | 12.5k | Acc |
| CARP (Zhang et al., 2023) | ✗ | ✗ | ✓ | Middle School | 4.9k | Acc |
| DynaMath (Zou et al., 2024) | ✗ | ✗ | ✗ | Mixed | 5010 | Acc+Rob |
| MathCheck (Zhou et al., 2024) | ✗ | ✗ | ✗ | Mixed | 4.5k | Acc+Rob |
| Omni-Math (Gao et al., 2024) | ✗ | ✗ | ✗ | Competition | 4.4k | Acc |
| FrontierMath (Glazer et al., 2024b) | ✓ | ✓ | ✗ | Research | Hundreds | Acc |
| HARP (Yue et al., 2024) | ✗ | ✗ | ✗ | Competition | 5.4k | Acc |
| OlympMATH (Sun et al., 2025) | ✗ | ✗ | ✗ | Competition | 200 | Acc |
| HARDMATH (Fan et al., 2024) | ✗ | ✗ | ✗ | Graduate | 1.5k | Acc |
| OlympiadBench (He et al., 2024) | ✗ | ✗ | ✗ | Competition | 8.5k | Acc |
| **MMRC (Ours)** | ✓ | ✓ | ✓ | **University** | **500** | **Acc+Eff** |

2024a) have been introduced to test advanced mathematical reasoning. In contrast, MMRC uniquely targets the integration of natural language and code execution, addressing tasks that require significant computational power. As illustrated in Table 1, MMRC offers several key advantages over existing benchmarks.

**Reasoning with Large Language Models.** The introduction of chain-of-thought (CoT) prompting (Wei et al., 2022a) marked a key advance in LLM reasoning by breaking down complex problems into intermediate steps (Sprague et al., 2024). Models like OpenAI-o1 (Jaech et al., 2024) and Deepseek-R1 (Guo et al., 2025a), along with reinforcement learning-based fine-tuning, have further enhanced large-scale reasoning. Ongoing research explores more efficient reasoning paradigms (Aggarwal & Welleck, 2025; Nayab et al., 2024; Lee et al., 2025) and self-evolving models (Shinn et al., 2023; Zelikman et al., 2022; Yang et al., 2023).

**Code-Integrated Mathematical Reasoning.** While LLMs show strong reasoning skills, they struggle with complex numerical computations, leading to inaccuracies and high computational costs. To address this, integrating code execution has become a key strategy. Initial approaches like Program of Thoughts (Chen et al., 2022) and MathCoder (Wang et al., 2023) focused on generating code for calculations. Later methods, including PAL (Gao et al., 2023), Tora (Gou et al., 2023), and Mumath-Code (Yin et al., 2024), combined natural language reasoning with code execution. However, these approaches have been primarily evaluated on easier datasets like GSM8K, where natural language is often sufficient. Our work builds on these agent-based methods and demonstrates that integrating code execution can significantly improve accuracy on more complex problems that truly require it.

## 3 THE MMRC BENCHMARK

### 3.1 OVERVIEW

The core objective of the MMRC dataset is to evaluate a model's ability to integrate mathematics with code in reasoning. It contains 500 carefully curated and adapted problems, each combining nontrivial logical structures with demanding computational requirements. To capture performance across different levels of difficulty, MMRC is divided into two subsets: MMRC-Base and MMRC-Hard.

Both subsets scale the computational component to challenging levels, involving combinatorial search spaces, deep recursion, or advanced numerical methods. The key distinction lies in their dependence on programmatic execution. Problems in MMRC-Base can, in principle, be solved manually but only through extremely tedious calculation, serving as a baseline for computational reasoning. In contrast, MMRC-Hard problems are effectively unsolvable without code, often requiring large-scale numerical simulation or recursive procedures. This two-tiered design enforces code-assisted reasoning while preserving the logical structure of mathematical derivations, requiring

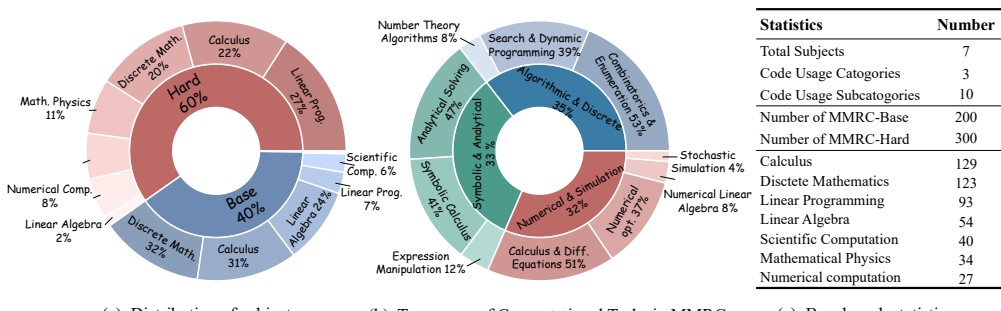

(a) Distribution of subjects     (b) Taxonomy of Computational Tasks in MMRC     (c) Benchmark statistics

Figure 2: **Statistical overview of the MMRC dataset.** (a) Distribution of mathematical subjects across the MMRC-Base (N=200) and MMRC-Hard (N=300) subsets. (b) Distribution of problems among the three major code usage categories. (c) Detailed statistics of the benchmark, including problem counts by subset and subject.

models to alternate between natural language reasoning and computational execution, an ability we consider essential for solving complex tasks.

**Subjects Coverage.** Unlike other benchmarks for complex mathematical problems that emphasize increasingly advanced knowledge beyond the practical scope of many models, MMRC is designed to evaluate general mathematical–code reasoning ability. Its coverage focuses on standard university-level mathematics while maintaining broad disciplinary diversity. As shown in Figure 2(a) and Figure 2(c), the dataset spans a wide range of subjects, including calculus, discrete mathematics, linear algebra, linear programming, mathematical physics, numerical computation, and scientific computing.

**Code Usage Coverage** Code execution enables LLMs to reduce reliance on natural language reasoning, lowering computational overhead while improving accuracy. For instance, computing the eigenvalues of a five-dimensional matrix is far more efficient than predicting them through token generation. MMRC tasks are therefore designed to test a hybrid reasoning-execution paradigm: reasoning and modeling are expressed in text, while final results are obtained through code execution. As shown in Figure 2(b), the code spans three major categories: (1) *Numerical and Simulation*, such as large-scale matrix operations and differential equation solving; (2) *Algorithmic and Discrete Methods*, including recursive search and dynamic programming; and (3) *Symbolic and Analytical Computation*, covering expression manipulation and equation solving. Detailed definitions and representative examples are provided in Appendix B.

## 3.2 DATA CURATION PIPELINE

**Data Collection and Rewriting.** To build a high-quality dataset, we collaborated with mathematics students from leading universities worldwide, primarily at the PhD and Master's level. They contributed to both problem creation and verification, covering domains such as calculus, linear algebra, probability theory, and operations research. To broaden coverage, experts from the natural sciences also provided domain-specific problems. Although MMRC does not aim to require highly advanced knowledge, strict accuracy and methodological rigor were enforced. Each contributor was assigned only to areas they had studied or applied within the past year, and all followed a structured workflow to organize and refine their problems.

**Collection.** The effectiveness of MMRC depends on the quality of its problems, each of which must satisfy two requirements. *Scalable Computational Complexity*: Problems should extend to large-scale tasks, not only by increasing numerical size but also by enriching computational logic through deeper recursion or more complex iterative structures. *Robust Reasoning Demand*: Problems must preserve substantial reasoning complexity, ensuring they challenge both natural language reasoning and computational procedures. This dual focus provides a rigorous test of a model's ability to perform code-assisted reasoning.

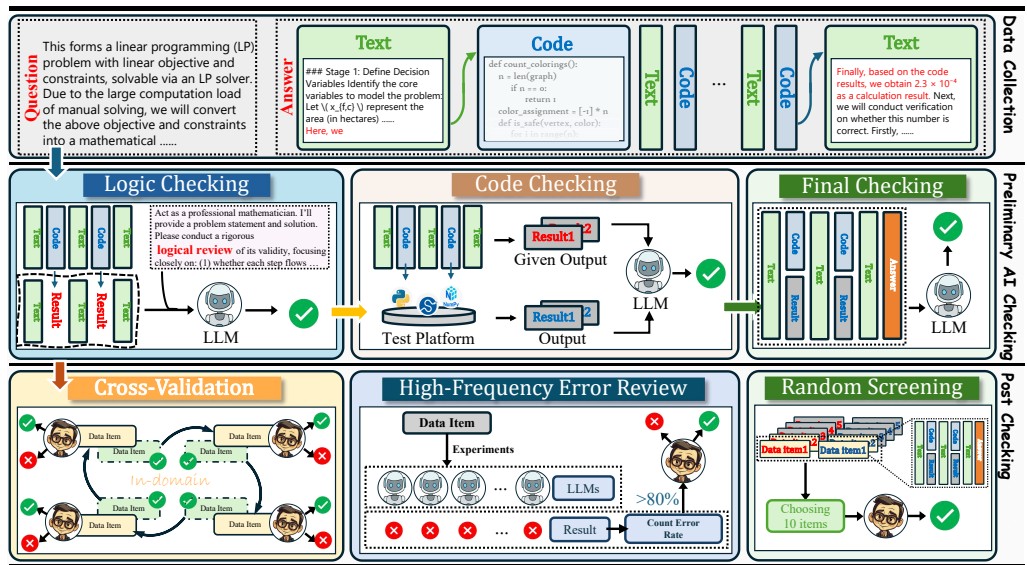

Figure 3: **Verification pipeline of the MMRC dataset.** The process consists of two stages: AI-based preliminary checking during data construction and human post-review after compilation. The preliminary stage includes three steps: logic consistency validation, code execution with result comparison, and a final acceptance decision made by a large language model. The post-review stage further ensures quality through cross-validation among contributors, re-examination of high-error problems, and random expert screening, where only subsets with all sampled problems passing are approved for release.

*Rewriting.* The original problems, though complex, did not fully meet our requirement that language-only reasoning should fail. We therefore adapted intermediate steps by replacing numerical data and computational logic with much larger values, substantially increasing complexity.

*Code Generation and Result Computation.* The expanded logic was intended to encourage solutions through executable code rather than long-form natural language reasoning. Contributors worked in a standardized Python environment with basic computational libraries. They generated executable code, embedded it into the problems, and computed the final solutions.

*Review and Submission.* Each problem underwent verification through code execution. Contributors submitted a complete package consisting of the original problem and solution, the adapted version, a description of the adaptation process, all code snippets, execution results, and source documentation.

**Data Verification.** Since MMRC problems are extensively adapted from their original sources, both reasoning logic and computational steps may diverge considerably. Ensuring correctness therefore requires a rigorous verification pipeline that goes beyond simple numerical checks. As illustrated in Figure 3, the process consists of two stages: a preliminary AI-based review during data construction and a comprehensive human review after dataset compilation.

*Preliminary AI Checking.* Each finalized problem undergoes a three-step automated check: *(1) Logic check*: Code outputs are substituted into the reasoning chain to confirm consistency. *(2) Code check:* Programs are executed in a standardized environment, and results are compared with the logic check while monitoring for errors. *(3) Final check:* A large model integrates these results to produce the acceptance decision.

*Post Human Review.* After compilation, problems are reviewed through three complementary mechanisms: *(1) Cross-validation:* Contributors in the same domain review each other's problems using the original solutions. *(2) High-error review:* Problems with high error rates in preliminary experiments are re-examined. *(3) Random Screening:* Senior experts review ten problems per domain, and only datasets where all sampled problems pass are approved for release.

### 3.3 Evaluation Framework

To rigorously evaluate model capabilities on the MMRC dataset, we designed a comprehensive evaluation framework. It assesses models across three distinct inference modes designed to probe different facets of their reasoning and tool-use abilities, coupled with a robust, two-stage verification protocol for accurate assessment.

Our framework probes model capabilities via three distinct modes: **Internal Reasoning (IR)**, for baseline tool-free problem-solving; **Code-Invoked Reasoning (CIR)**, for generating a complete solution as a single program; and **Math-Code Agent (MCA)**, for iterative problem-solving with a code interpreter. The critical difference between the two tool-assisted modes is that CIR tests a model's ability for comprehensive, "one-shot" programmatic planning, whereas MCA evaluates its capacity for interactive reasoning and step-by-step refinement based on execution feedback.

To handle the diverse answer formats in MMRC, our two-stage verification protocol first uses a powerful LLM judge for efficient initial triage. Cases marked as `Uncertain` are then passed to a second-stage formal verifier, which uses a symbolic mathematics library for a definitive and precise judgment. A detailed exposition of this framework, including the evaluation modes, verification protocol, prompts, and environment setup, is provided in Appendix C.

## 4 Experiments

To systematically evaluate the ability of modern LLMs to integrate reasoning with code, we conducted a comprehensive set of experiments on the MMRC benchmark. These experiments address three central questions: how accuracy varies across models and reasoning paradigms when comparing analytically tractable problems in the base set with code-dependent problems in the hard set; what computational costs, measured in token consumption, are associated with different solution strategies and how these costs relate to performance; and what predominant failure modes arise when models attempt to solve complex problems, revealing the key bottlenecks in current reasoning and coding abilities.

### 4.1 Experimental Setup

**Evaluation Protocol.** Our evaluation spans the base and hard subsets, testing models under three inference modes (IR, CIR, and MCA) and two prompting styles inspired by cognitive frameworks (Kahneman, 2011; Wei et al., 2022b). Fast models directly generate answers in a single pass (e.g., DEEPSEEK V3), while slow models produce explicit reasoning or plans before giving final solutions (e.g., DEEPSEEK R1). We evaluate 20 representative LLMs, including proprietary, open-source, and math/code-specialized models. The full model list with versions, API endpoints, and other information is provided in Appendix D.

**Execution and Decoding.** All experiments used greedy decoding (temperature 0) with a maximum context of 128k tokens and a per-turn output limit of 60k tokens. MCA mode was capped at 10 iterations. For coding-augmented modes (CIR and MCA), code was executed in a secure Python sandbox with a 30-second timeout and 512MB memory limit. Additional details on the environment and prompts are provided in Appendix C.3 and Appendix C.4.

**Overview of Analysis.** Table 2 summarizes the overall evaluation results. The following sections examine these findings in relation to our research questions: §4.2 analyzes accuracy trends, §4.3 evaluates efficiency, and §4.4 provides an error analysis.

### 4.2 Main Results: Accuracy Analysis

The accuracy results in Table 2 and Figure 4 highlight three main findings on integrated mathematical reasoning.

**MMRC validates the necessity of code execution.** A sharp performance drop from the base to the hard set confirms the benchmark design. For example, DEEPSEEK V3.1-FAST falls from 84.4% (IR, base) to 24.1% (IR, hard). Even the best hard-set accuracy, 43.4% by CLAUDE 4 SONNET-FAST with CIR, remains modest, underscoring the challenge of robustly combining reasoning with code.

Table 2: **Main Results on Base and Hard Questions. Bold numbers** indicate the best performance within each column

| Metric / Model | IR Base ACC | IR Base Tok | IR Hard ACC | IR Hard Tok | CIR Base ACC | CIR Base Tok | CIR Hard ACC | CIR Hard Tok | MCA Base ACC | MCA Base Tok | MCA Hard ACC | MCA Hard Tok |
|---|---|---|---|---|---|---|---|---|---|---|---|---|
| *Open-source Models* | | | | | | | | | | | | |
| DeepSeek R1-slow | 61.3% | 3.8k | 0.5% | **0.4k** | 81.1% | 5.1k | 29.7% | 6.0k | 76.9% | 4.7k | 13.7% | 5.3k |
| | - | - | - | - | +19.8% | +1.2k | +29.2% | +5.6k | +15.6% | +0.9k | +13.2% | +5.0k |
| DeepSeek V3-fast | 73.6% | 1.1k | 15.6% | 2.1k | 70.3% | 0.3k | 25.9% | 0.5k | 75.5% | 1.0k | 29.7% | 1.9k |
| | - | - | - | - | -3.3% | -0.8k | +10.4% | -1.5k | +1.9% | -0.1k | +14.2% | -0.1k |
| DeepSeek V3.1-fast | **84.4%** | 1.3k | 24.1% | 2.9k | 72.6% | 0.4k | 30.2% | 0.7k | **87.7%** | 1.6k | **40.1%** | 3.6k |
| | - | - | - | - | -11.8% | -0.9k | +6.1% | -2.2k | +3.3% | +0.3k | +16.0% | +0.7k |
| GLM-4.5 Air-fast | 74.1% | 1.7k | 10.8% | 10.6k | 63.7% | 0.4k | 18.9% | 1.0k | 75.5% | 2.9k | 26.9% | 6.4k |
| | - | - | - | - | -10.4% | -1.3k | +8.0% | -9.5k | +1.4% | +1.2k | +16.0% | -4.1k |
| GLM-4.5 Air-slow | 76.9% | 13.0k | 22.6% | 40.8k | 81.1% | 33.7k | 26.9% | 41.8k | 78.3% | 37.7k | 18.4% | 50.3k |
| | - | - | - | - | +4.2% | +20.6k | +4.2% | +1.0k | +1.4% | +24.7k | -4.2% | +9.5k |
| LLaMA 4 Scout-fast | 61.3% | 1.0k | 6.6% | 1.2k | 66.5% | 0.3k | 15.6% | 0.5k | 70.8% | 1.1k | 16.0% | 1.8k |
| | - | - | - | - | +5.2% | -0.6k | +9.0% | -0.7k | +9.4% | +0.1k | +9.4% | +0.6k |
| Qwen3 235B-fast | 74.5% | 1.5k | 10.8% | 2.5k | 71.2% | **0.3k** | 22.2% | **0.5k** | 76.4% | 1.1k | 29.7% | 2.5k |
| | - | - | - | - | -3.3% | -1.2k | +11.3% | -2.0k | +1.9% | -0.4k | +18.9% | -0.0k |
| Qwen3 235B-slow | 83.0% | 14.5k | 29.2% | 28.0k | 81.6% | 14.7k | 36.8% | 20.7k | 79.7% | 14.8k | 29.7% | 26.2k |
| | - | - | - | - | -1.4% | +0.2k | +7.5% | -7.4k | -3.3% | +0.3k | +0.5% | -1.9k |
| Qwen3 32B-fast | 73.1% | 2.9k | 8.5% | 12.9k | 26.9% | 17.0k | 9.0% | 10.4k | 73.6% | 1.3k | 24.5% | 3.8k |
| | - | - | - | - | -46.2% | +14.2k | +0.5% | -2.5k | +0.5% | -1.6k | +16.0% | -9.1k |
| Qwen3 32B-slow | 80.7% | 16.8k | 22.2% | 45.8k | 35.4% | 23.9k | 23.1% | 29.0k | 75.9% | 16.1k | 22.6% | 41.6k |
| | - | - | - | - | -45.3% | +7.0k | +0.9% | -16.8k | -4.7% | -0.8k | +0.5% | -4.1k |
| Qwen3 8B-fast | 64.6% | 1.8k | 6.6% | 8.9k | 54.7% | 0.6k | 12.7% | 1.6k | 68.4% | 1.0k | 15.1% | 3.5k |
| | - | - | - | - | -9.9% | -1.2k | +6.1% | -7.3k | +3.8% | -0.8k | +8.5% | -5.4k |
| Qwen3 8B-slow | 80.2% | 22.9k | 18.9% | 49.0k | 76.9% | 14.7k | 29.2% | 29.9k | 78.8% | 17.6k | 19.8% | 45.8k |
| | - | - | - | - | -3.3% | -8.2k | +10.4% | -19.1k | -1.4% | -5.3k | +0.9% | -3.1k |
| *Proprietary Models* | | | | | | | | | | | | |
| Claude 4 Sonnet-fast | 83.0% | 1.1k | 17.0% | 1.3k | **84.4%** | 0.8k | **43.4%** | 1.2k | 83.5% | 3.7k | 34.0% | 3.7k |
| | - | - | - | - | +1.4% | -0.3k | +26.4% | -0.1k | +0.5% | +2.6k | +17.0% | +2.4k |
| Gemini 2.5 Flash-slow | 81.6% | 5.5k | 20.3% | 19.7k | 79.2% | 3.6k | 34.9% | 14.3k | 80.7% | 6.5k | 31.1% | 20.8k |
| | - | - | - | - | -2.4% | -1.9k | +14.6% | -5.4k | -0.9% | +1.0k | +10.8% | +1.1k |
| Gemini 2.5 Flash Lite-fast | 84.4% | 6.9k | 24.5% | 31.9k | 78.8% | 6.3k | 30.7% | 16.6k | 85.4% | 5.7k | 24.1% | 24.5k |
| | - | - | - | - | -5.7% | -0.6k | +6.1% | -15.4k | +0.9% | -1.3k | -0.5% | -7.4k |
| Gemini 2.5 Pro-slow | 82.5% | 5.7k | **33.0%** | 14.9k | 71.2% | 5.5k | 32.5% | 11.8k | 70.8% | 6.2k | 36.8% | 12.1k |
| | - | - | - | - | -11.3% | -0.2k | -0.5% | -3.1k | -11.8% | +0.6k | +3.8% | -2.8k |
| *Specialized Models* | | | | | | | | | | | | |
| Mathstral 7B-fast | 27.4% | 0.9k | 0.5% | 1.2k | 29.2% | 0.3k | 1.4% | 0.5k | 32.5% | 1.5k | 5.7% | 1.9k |
| | - | - | - | - | +1.9% | -0.6k | +0.9% | -0.7k | +5.2% | +0.5k | +5.2% | +0.7k |
| Qwen2.5-Math 72B-fast | 56.6% | 1.0k | 4.7% | 1.1k | 62.7% | 0.8k | 11.8% | 0.9k | 59.0% | 0.9k | 3.3% | 1.1k |
| | - | - | - | - | +6.1% | -0.2k | +7.1% | -0.2k | +2.4% | -0.0k | -1.4% | -0.0k |
| Qwen2.5-Math 7B-fast | 49.1% | **0.8k** | 2.4% | 1.0k | 57.1% | 0.6k | 8.5% | 0.8k | 47.6% | **0.9k** | 2.4% | **1.1k** |
| | - | - | - | - | +8.0% | -0.2k | +6.1% | -0.3k | -1.4% | +0.1k | +0.0% | +0.1k |
| Qwen3-Coder 30B-fast | 55.7% | 2.2k | 7.5% | 4.2k | 73.6% | 0.7k | 22.6% | 3.6k | 63.7% | 3.0k | 21.2% | 4.7k |
| | - | - | - | - | +17.9% | -1.5k | +15.1% | -0.6k | +8.0% | +0.8k | +13.7% | +0.4k |

**Code augmentation is not universally beneficial.** On the base set, where problems remain analytically tractable, forcing code often reduces accuracy. QWEN3-32B-SLOW drops from 80.7% (IR) to 35.4% (CIR). MCA mode is more stable, as its flexibility helps avoid unnecessary tool calls, showing that code should be used selectively.

**On hard problems, coding is indispensable.** For the hard set, both CIR and MCA consistently outperform IR. CIR often provides reliable gains (e.g., QWEN3-235B-SLOW: 29.2% IR → 36.8% CIR). The best overall result is also achieved under the CIR mode, with CLAUDE 4 SONNET-FAST reaching 43.4%. Further analysis of CIR and MCA modes, as well as the impact of the fast vs. slow setting, is provided in Appendix E.

## 4.3 ANALYSIS OF COMPUTATIONAL COST

Beyond accuracy, computational cost measured by token consumption is a critical factor for assessing the practical viability of different problem-solving strategies. Our analysis based on the Tok data in Table 2 shows that the relationship between cost and performance is nuanced, revealing both opportunities for efficiency and risks of inefficiency.

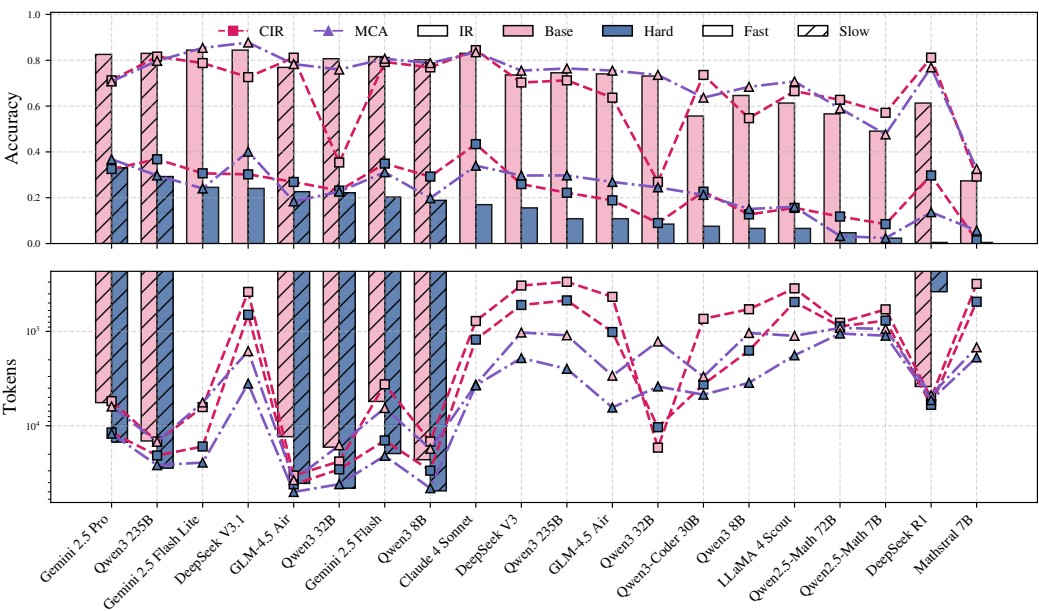

Figure 4: **Overall accuracy on the MMRC base and hard subsets.** This figure provides a high-level comparison of model performance across different settings, highlighting the stark difficulty increase on the hard set.

**Code as a reasoning compressor on the base set.** Tool-augmented modes often consume fewer tokens than pure IR because analytically tractable problems still require lengthy step-by-step derivations in text. Code provides a compact representation. This effect is most evident in slow models with high IR costs. For example, QWEN3-32B-SLOW reduces its usage from 45.8k (IR) to 29.0k (CIR), saving more than 15k tokens.

**Efficiency of MCA vs. CIR depends on model style.** Neither tool-augmented mode is consistently more efficient. Fast models often benefit from MCA due to concise tool calls, while slow models gain more from CIR, which replaces long reasoning traces with a single script. For instance, DEEPSEEK V3.1-FAST saves 2.2k tokens in CIR compared to IR, whereas MCA slightly increases cost.

**The efficiency trap.** Higher token consumption does not guarantee higher accuracy, particularly on the hard set. MCA is prone to unproductive iterative loops. For example, GLM-4.5 AIR-SLOW achieves 26.9% accuracy with 41.8k tokens in CIR, but only 28.4% with 50.3k tokens in MCA. This shows that efficient and well-structured strategies outperform verbose ones prone to reasoning drift.

### 4.4 ERROR ANALYSIS

The distribution of errors is highly model-dependent, reflecting differences in architecture and training. Some models perform well in CIR by generating correct programs in one pass but struggle with iterative MCA refinement, while others adapt better interactively yet fail at producing complex monolithic scripts. In the main analysis, we focus on GEMINI-2.5-PRO, whose strong overall accuracy makes its failure modes particularly informative. To diagnose errors systematically, we map them onto a reasoning-to-code pipeline: (1) *Understanding*—grasping the problem and constraints; (2) *Reasoning*—formulating the correct abstract plan and mathematical model; (3) *Implementation*—translating the plan into correct code logic; (4) *Syntax*—producing syntactically valid programs; and (5) *Format*—adhering to output specifications. This framework enables us to locate failures within specific cognitive stages rather than merely counting them, with detailed examples provided in Appendix G.

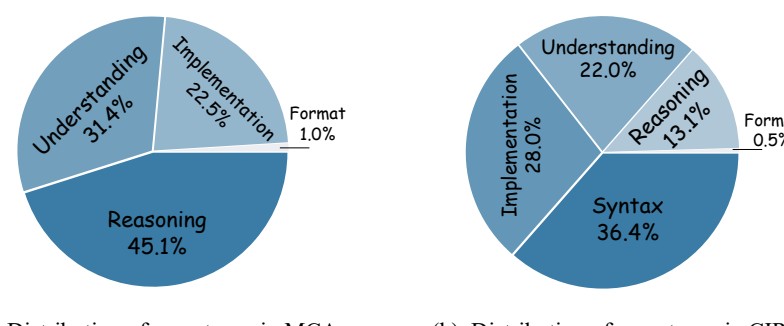

(a) Distribution of error types in MCA       (b) Distribution of error types in CIR

Figure 5: **Distribution of error types under the MCA and CIR modes for GEMINI-2.5-PRO.** The error profiles differ dramatically, highlighting how each mode tests different aspects of the model's capabilities.

**Analysis of errors inherent to the method**    Our error analysis reveals a clear contrast between the two execution modes. In the CIR setting, where the model must deliver a full solution in a single attempt, failures are dominated by **Syntax** (36.4%) and **Implementation** (28.0%) errors, highlighting the brittleness of monolithic planning: a single slip at the coding level derails the entire solution. This reflects the unrealistic demand that the model simultaneously act as both a precise mathematician and a flawless programmer. In contrast, the MCA mode, which allows iterative interaction with an interpreter, almost eliminates syntax-level issues (1.0%) by leveraging feedback for local corrections. Yet this advantage reveals a deeper limitation: errors shift predominantly to **Reasoning** (45.1%) and **Understanding** (31.4%), showing that even when surface-level coding problems are resolved, the model often constructs fundamentally flawed plans. Taken together, these results suggest that CIR exposes fragility in execution, while MCA exposes fragility in reasoning, offering complementary perspectives on model limitations.

**Comparison between Methods**    When comparing error sources between MCA and CIR, we find that **Understanding** is a major failure mode in both settings, with comparable proportions, indicating that misinterpretation of the problem statement almost always invalidates subsequent reasoning. A similar pattern holds for **Implementation** errors. However, the two modes diverge sharply in coding-related failures: while MCA exhibits almost no code errors, CIR shows a high incidence of such mistakes (dominated by the **Syntax** category in Figure 5), reinforcing that interactive agent–environment feedback is critical for reliable task completion and that single-shot execution remains insufficient. Moreover, the proportion of **Reasoning** errors is higher in MCA than in CIR, often because the model *over-suspiciously* questions correct intermediate code results, reflecting a lack of training in effectively leveraging program outputs. Taken together, this highlights a central dilemma in math-code reasoning: CIR is brittle due to coding failures, while MCA exposes deeper reasoning flaws. Simply adding tools is insufficient—poor reasoning fails regardless of coding, and unreliable coding prevents sound reasoning from being executed. Progress therefore hinges on improving both intrinsic reasoning and reasoning-in-the-loop, aiming to build models that integrate robust reasoning with reliable coding.

## 5 CONCLUSION

In this work, we introduced MMRC, a benchmark of 500 curated problems designed to evaluate the integration of mathematical reasoning and code execution in large language models. By focusing on university-level mathematics and deliberately imposing substantial computational workloads, MMRC moves beyond text-only benchmarks and enforces the use of executable code for accurate solutions. Our evaluation of nearly 120 open- and closed-source models under both program-aided and agent-based paradigms shows that code integration consistently improves performance on complex tasks while reducing token usage. These results establish MMRC as the first systematic benchmark for math–code integrated reasoning and highlight its value in advancing LLMs toward real-world problem solving. Looking ahead, MMRC opens opportunities for developing models that can more seamlessly combine symbolic reasoning with computational execution, paving the way for LLMs to tackle increasingly sophisticated tasks in science, engineering, and beyond.

ETHICS STATEMENT

Our work primarily introduces and analyzes a new dataset for evaluating how large language models (LLMs), in combination with programmatic tools, can tackle complex mathematical reasoning tasks. While our emphasis is on dataset design and evaluation protocols rather than human data collection, we acknowledge the following ethical considerations and limitations:

- **Model licensing and usage compliance.** Some of the evaluated LLMs are proprietary, with usage governed by license agreements. We ensure that all API calls and usage adhere to the providers' terms of service. All prompt templates, wrapper code, and evaluation scripts we release respect these licenses and do not reverse engineer or redistribute proprietary weights.
- **Data provenance and reuse.** Our MMRC benchmark is derived from existing mathematical reasoning sources (e.g. textbooks, problem sets) via human adaptation. We verify that the original sources allow non-commercial reuse or are in public domain. We also screen for inadvertent overlaps with model training data (contamination) and remove suspicious items from evaluation.
- **Misuse potential.** Since our work focuses on releasing a benchmark dataset rather than an interactive solving system, we do not anticipate direct risks of enabling academic dishonesty (e.g., homework or exam solving). The dataset and accompanying code are intended strictly for research purposes, and we encourage responsible use within the scientific community.

REPRODUCIBILITY STATEMENT

We are committed to ensuring the reproducibility of our research. To support the verification and extension of our findings, we provide comprehensive details of our experimental methodology in the main paper and appendices.

- **Experimental Setup:** We provide a detailed description of our experimental setup, hyperparameters, and evaluation protocols in Section 4 and Appendix C.
- **Model Specifications:** Key details of all evaluated models, including their exact versions and configurations, are thoroughly documented in Appendix D.
- **Data and Code Availability:** The MMRC benchmark dataset and the source code used to generate the results in this paper will be made publicly available upon publication.

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

## A  LLM USAGE

Large Language Models (LLMs) were utilized in several capacities in this work. Primarily, LLMs are the subject of our study, as we evaluate their capabilities using the MMRC benchmark. An LLM was also used as a component in our evaluation methodology to perform an initial triage of answers, and as an assistant for the linguistic polishing of this manuscript, including tasks like grammar checking and improving clarity.

The core scientific work, including the ideation and design of the MMRC benchmark, the experimental methodology, and the analysis of results, was conducted entirely by the human authors. The LLM's role was strictly limited to the functions described above, with no contribution to the research ideas or findings. The authors take full responsibility for all content in this paper and have verified the accuracy of any text assisted by the LLM.

## B  CODE USAGE DETAILS

Table 3: **Taxonomy of Computational Methods in MMRC Problems.** This table outlines the classification system used to analyze Python solutions, categorized by their core computational paradigm.

| Code | Category | Description | Typical Libraries & Functions |
|---|---|---|---|
| **Major Category 1: Numerical & Simulation (NS)** | | | |
| *Methods employing numerical computation and simulation techniques for approximate solutions.* | | | |
| NS-LA | Numerical Linear Algebra | Solving systems of linear equations, eigenvalue problems, or other large-scale matrix operations. | `numpy.linalg.solve`, `numpy.linalg.eig` |
| NS-OPT | Numerical Optimization | Finding the minimum or maximum of a continuous function, including non-linear optimization and linear programming. | `scipy.optimize.minimize`, `scipy.optimize.linprog` |
| NS-CDE | Calculus & Diff. Equations | Numerically computing integrals, derivatives, roots, or solving ordinary differential equations (ODEs). | `scipy.integrate.quad`, `scipy.integrate.solve_ivp` |
| NS-SIM | Stochastic Simulation | Using random sampling to model stochastic processes or to estimate numerical quantities (e.g., Monte Carlo methods). | `numpy.random`, `random`, `statistics` |
| **Major Category 2: Algorithmic & Discrete (AD)** | | | |
| *Methods for problems in discrete domains, focusing on exact solutions via structured exploration.* | | | |
| AD-CMB | Combinatorics & Enumeration | Counting or generating permutations, combinations, or subsets by systematically iterating through discrete solution spaces. | `itertools`, `math.comb`, `math.perm` |
| AD-SDP | Search & Dynamic Programming | Algorithms based on recursion, search (e.g., tree/graph traversal), or dynamic programming to find solutions. | custom recursive functions, `functools.lru_cache` |
| AD-NT | Number Theory Algorithms | Problems involving prime numbers, modular arithmetic, greatest common divisors, or other number-theoretic properties. | `math.gcd`, `pow(a,b,m)` |
| **Major Category 3: Symbolic & Analytical (SA)** | | | |
| *Methods that manipulate mathematical expressions to find exact, symbolic solutions.* | | | |
| SA-MAN | Expression Manipulation | Simplifying, expanding, factoring, or transforming mathematical expressions without numerical evaluation. | `sympy.simplify`, `sympy.expand` |
| SA-SLV | Analytical Solving | Finding exact symbolic solutions to equations, systems of equations, or differential equations. | `sympy.solve`, `sympy.dsolve` |
| SA-CAL | Symbolic Calculus | Computing derivatives, integrals, limits, or series expansions symbolically rather than numerically. | `sympy.diff`, `sympy.integrate`, `sympy.limit` |

This appendix provides a detailed breakdown of the code usage taxonomy developed for the MMRC benchmark. As introduced in the main text, a core feature of MMRC is its requirement for models to master a diverse range of computational tasks using code. To facilitate a systematic and fine-grained analysis of these requirements, we organized the problems into a three-tiered classification system based on their core computational paradigm.

The following Table 3 presents this taxonomy in full detail. It expands upon the three major categories introduced previously—Numerical & Simulation (NS), Algorithmic & Discrete (AD), and Symbolic & Analytical (SA)—by breaking them down into ten distinct subcategories. For each subcategory, the table provides its definition, a unique code, and a list of representative Python libraries and functions typically required for its solution.

## C  EVALUATION DETAILS

This appendix provides a detailed exposition of the evaluation framework used for the MMRC benchmark, as introduced in Section 3.3. The overall process, from inference modes to the two-stage verification protocol, is illustrated in Figure 6.

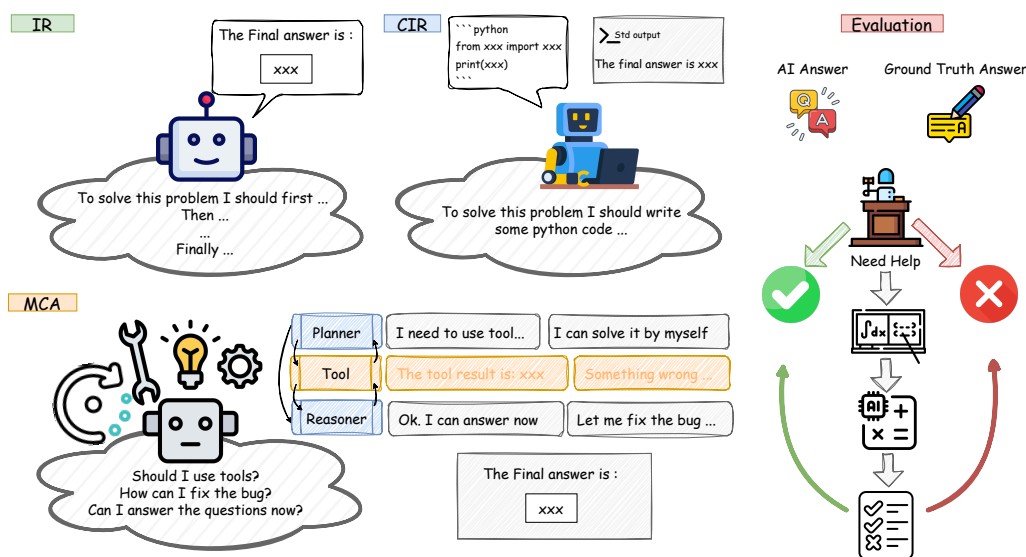

Figure 6: **Illustration of the complete inference and evaluation framework for the MMRC benchmark.** It depicts the three distinct inference modes (IR, CIR, MCA) and the subsequent two-stage answer verification protocol.

## C.1 INFERENCE MODES

To provide a holistic view of model performance, we evaluate each model under three inference modes, each designed to probe a specific aspect of a model's reasoning and coding capabilities.

**Internal Reasoning (IR).** This mode measures a model's intrinsic mathematical reasoning without external computational aid. It serves as a baseline for a model's capacity for internal calculation and logical deduction. Models are prompted using the Chain-of-Thought paradigm to "think step by step" and articulate their intermediate reasoning in natural language (Wei et al., 2022b). For standardized answer extraction, the model must present its final answer enclosed in a \boxed{} environment, a common convention in mathematical benchmarks (Hendrycks et al., 2021; Sun et al., 2024).

**Code-Invoked Reasoning (CIR).** This mode assesses the model's ability to structure a complete programmatic solution and offload computation to an external tool. Inspired by the Program-Aided Language (PAL) approach (Gao et al., 2023), the model is prompted to translate its entire solution into a single, self-contained Python script, embedding its reasoning as comments. The generated script is then executed by a Python interpreter and must print the final answer to standard output in the format `Final Answer: <result>`.

**Math-Code Agent (MCA).** The DE mode evaluates iterative problem-solving and the dynamic use of coding tools (Yan et al., 2024). In our setup, the model is connected to a Python interpreter and operates in a loop of reasoning, optional code execution, and observation. It acts as both a **reasoner** and a **planner** (Yan et al., 2024), deciding when to invoke code, which libraries to use, and how to incorporate execution results into subsequent reasoning steps. The model may perform multiple cycles of thought, action (code execution), and observation to correct errors and refine its approach, which is critical for multi-step computational tasks.

To ensure a fair comparison, both PS and DE modes run within an identical, secure Python sandbox.

## C.2 TWO-STAGE ANSWER VERIFICATION PROTOCOL

Given the diverse nature of answers in MMRC, which can range from single numerical values to complex expressions or sets, a simple string match is insufficient for accurate evaluation. We therefore employ a robust, two-stage evaluation protocol that combines the efficiency of an LLM judge with the precision of a formal verifier, as depicted in Figure 6.

**Stage 1: LLM Judge for Initial Triage.** In the first stage, we utilize a powerful LLM judge (e.g., GPT-4o) to perform an initial assessment. The judge is provided with the problem description, the model's extracted answer, and the ground-truth answer. Its task is to classify the response into one of three categories: `Correct`, `Incorrect`, or `Uncertain`. For straightforward cases involving simple numerical or textual comparisons, the LLM judge can often make a definitive ruling.

**Stage 2: Formal Verification for Ambiguous Cases.** When the LLM judge outputs `Uncertain`, it typically signifies a scenario requiring formal verification. This often occurs when the model's output is a complex symbolic expression while the ground truth is a numerical value (or vice-versa). In such cases, the LLM judge is further prompted to extract the model's answer as a formal LaTeX expression. This expression is then passed to a second-stage verification module. This module, inspired by the methodology of the Math-Verify project (Kydlíček), uses a symbolic mathematics library (e.g., SymPy) to compute the exact value of the expression. The computed value is then compared against the ground-truth answer with a predefined tolerance to make a final, authoritative judgment.

## C.3 PYTHON SANDBOX ENVIRONMENT

All programmatic computations used by the benchmark (numerical calculations, linear algebra, optimization, and symbolic operations) are executed inside a controlled Python sandbox. The sandbox is exposed via a single programmatic interface.

---

**Python Executor Interface**

**Function signature:**

```
python_executor(query: str) -> str
```

The function accepts a Python source string `query`, executes it in an isolated process, and returns the captured `stdout` as a UTF-8 string. To standardize output parsing, user code is expected to explicitly emit final results using `print()`. If execution fails, the returned string contains the full Python traceback (with error type and the failing line number relative to the supplied `query`), facilitating automated debugging.

**A canonical example call is:**

```
code = "import math\nprint(f'sqrt(25) = {math.sqrt(25)}')"
out = python_executor(code)
# out == "sqrt(25) = 5.0\n"
```

**If an error occurs, the returned string contains the traceback (example format):**

```
Traceback (most recent call last):
  File "<string>", line 3, in <module>
ZeroDivisionError: division by zero
```

---

**Supported / curated libraries.** For the automated agents and evaluation harness we restrict and endorse a compact scientific stack. The core approved libraries and their versions used in experiments are shown in Table 4.

**Execution constraints and safety.** To ensure deterministic, safe, and reproducible runs we enforce the following runtime constraints for each independent execution of `python_executor`:

- **Wall-clock time limit:** 30 seconds per call (hard timeout).
- **Memory limit:** 512 MB resident memory per call.
- **Network access:** disabled for sandboxed executions (no external HTTP/DNS/socket access).
- **Filesystem:** writable ephemeral working directory per execution; persistent write access is disallowed. The ephemeral directory is purged on process termination.
- **Process isolation:** executions run in short-lived isolated processes (container or restricted interpreter) with system-call filtering and limited privileges.

Timeouts, memory overflows, or prohibited operations result in an immediate termination and a returned error string describing the failure mode.

Table 4: **Core libraries available to the sandboxed executor environment.** Complete package specifications are detailed in the supplementary materials.

| Library | Role | Version |
|---------|------|---------|
| **Third-Party Libraries** | | |
| numpy | Arrays and numerical computations | 2.2.6 |
| scipy | Optimization, integration, linear algebra, statistics | 1.16.1 |
| pandas | Tabular data manipulation and analysis | 2.3.2 |
| sympy | Symbolic mathematics computation | 1.14.0 |
| **Standard Libraries** | | |
| math | Basic mathematical functions (real numbers) | Standard Library |
| cmath | Mathematical functions for complex numbers | Standard Library |
| random | Generate pseudo-random numbers | Standard Library |
| statistics | Mathematical statistics functions | Standard Library |
| itertools | Functions creating iterators for efficient looping | Standard Library |
| functools | Higher-order functions and operations on callable objects | Standard Library |
| json | JSON encoder and decoder | Standard Library |
| re | Regular expression operations | Standard Library |

## C.4 PROMPTS

In this section, we present the prompt templates that orchestrate the three inference modes, along with the judge prompt that evaluates student answers against references. We also describe the.

---

**Tool: `python_executor`**

**Description:**
Executes Python code in a secure sandboxed environment for mathematical and scientific computations.
This is your primary tool for any task requiring precise numerical computation or data analysis. You can use this tool for problems involving:

- Mathematical calculations (calculus, algebra, statistics, number theory)
- Linear algebra (matrices, eigenvalues, determinants, decompositions)
- Optimization problems (linear programming, nonlinear optimization)
- Scientific computing (numerical integration, differential equations)
- Data analysis and manipulation
- Complex symbolic mathematics
- Any multi-step calculation that requires precision

**IMPORTANT:** Always include `print()` statements to display results. The tool captures both successful outputs and detailed error information with line numbers for debugging.

**Args:**
`query`: A string of valid Python code to execute. Must include `print()` statements for output. The code must be syntactically correct Python and use only the allowed libraries.

**Available Libraries:**

- **Core:** `math`, `cmath`, `random`, `statistics`, `itertools`, `functools`, `json`, `re`
- **Arrays & Matrices:** `numpy` (as `np`), all numpy functions
- **Scientific:** `scipy` (`optimize`, `integrate`, `linalg`, `special`, `stats`), all scipy functions
- **Symbolic:** `sympy`, all sympy functions (`symbols`, `solve`, `integrate`, `diff`, etc.)
- **Data:** `pandas` (as `pd`), `collections`
- **Math constants:** `math.pi`, `math.e`, numpy constants

**Returns:**
`str`: Program output or detailed error information with line numbers for debugging. Errors include the specific line number and error type to help you fix issues.

**Examples:**

- **Basic math:** `python_executor("import math; print(f'sqrt(25) = {math.sqrt(25)}')")`
- **Linear algebra:** `python_executor("import numpy as np; A = np.array([[1,2],[3,4]]); print('det(A) =', np.linalg.det(A))")`
- **Optimization:** `python_executor("from scipy.optimize import minimize; result = minimize(lambda x: x[0]**2, [1]); print('minimum at x =', result.x[0])")`
- **Symbolic:** `python_executor("from sympy import symbols, solve; x = symbols('x'); sol = solve(x**2 - 4, x); print('Solutions:', sol)")`

## Internal Reasoning (IR) Mode

**System Prompt:**
You are an expert mathematician.
Guidelines:

- Read the problem carefully and identify key mathematical concepts
- Show your work step by step with clear reasoning
- Be precise and accurate in calculations
- End with EXACTLY ONE \boxed{final_answer} on the last line

**User Prompt:**
Problem:
{question}
Please solve this problem step by step and end with EXACTLY ONE \boxed{final_answer} on the last line.

## Math Code Agent (MCA) Mode

**System Prompt:**
You are an expert mathematician who can execute Python code via the tool python_executor.
Guidelines:

- Read the problem carefully and identify key mathematical concepts
- Show your work step by step with clear reasoning
- If nontrivial computation is needed (calculus, equation solving, matrix ops, statistics, long arithmetic), FIRST write Python and call python_executor. Do NOT fabricate results.
- Answer format: end with EXACTLY ONE \boxed{final_answer} on the last line.

**User Prompt:**
Problem:
{question}
Please solve this problem step by step and end with EXACTLY ONE \boxed{final_answer} on the last line.

**Code-Invoked Reasoning (CIR) Mode**

**System Prompt:**
You are an expert mathematician who solves problems by writing Python code.
**Your Task:**
Instead of solving the problem directly, write a Python program that will solve the problem for you. Your program should:

- Read and understand the mathematical problem
- Implement the solution logic in Python
- Compute the answer using mathematical calculations
- Output the final answer in the specified format

**Available Libraries:**
Core: math, cmath, random, statistics, itertools, functools, json, re
Arrays & Matrices: numpy (as np), all numpy functions
Scientific: scipy (optimize, integrate, linalg, special, stats), all scipy functions
Symbolic: sympy, all sympy functions (symbols, solve, integrate, diff, etc.)
Data: pandas (as pd), collections
Math constants: math.pi, math.e, numpy constants
**Code Format:**
Write your Python code inside ```python and ``` blocks.
**Important Guidelines:**

- Write complete, executable Python code
- Include all necessary imports
- Add comments to explain your approach
- ALWAYS end with: `print(``Final Answer:'', your_result)`
- Ensure your code is mathematically correct and handles edge cases
- Do NOT write any explanation outside the code block

**Output Requirements:**

- Write ONLY the Python code block, no additional text
- The code should be self-contained and executable
- The final line must print: "Final Answer: [your calculated result]"

**User Prompt:**
Problem:
{question}
Write a Python program to solve this problem. Your program must end with:
`print(``Final Answer:'', your_result)`

### Judge / Answer-Matching

**System Prompt:**

You will receive two strings: `predicted_answer` (student) and `correct_answer` (reference). Your goal is to decide whether the student's *final* answers match the reference.

**Guidelines**

- **Extract** only the student's final answers from `predicted_answer`; ignore intermediate steps.
- Judge **mathematical equivalence only**.

**Defer-to-verify policy**

- *Only* if a clear decision cannot be made but could be settled by simple symbolic/numeric checks, set:
  - `"needs_math_verify": true`
  - `"is_correct": false` (decision deferred)
  - and provide `"verify_expressions_latex"`.
- Otherwise (clear match/mismatch), set `"needs_math_verify": false` and give final `"is_correct"` with confidence.

**STRICT `verify_expressions_latex` (when `needs_math_verify=true`)**

- Canonicalize every item into valid LaTeX *math* and wrap with $$...$$ (numbers too).
- Auto-convert: use $\sin, \cos, \log, \ln, \exp$, constants $e, \pi, i$.
- Exponential policy: use $\exp(x)$ for exponentials; use $e$ only when the constant stands alone.
- Non-standard multi-letter identifiers: $\operatorname{...}$.
- Provide an object with **equal-length** lists in 1–1 correspondence:

```
{
"ground_truth":  ["$$...$$", "$$...$$", ...],
"student":       ["$$...$$", "$$...$$", ...]
}
```

**Return format (JSON only)**

- `"is_correct"`: boolean,  `"confidence"`: float in [0,1]
- `"reasoning"`: brief string
- `"needs_math_verify"`: boolean
- `"verify_expressions_latex"`:     object    above    (only    if `needs_math_verify=true`)

**User Prompt:**

Problem: {question}

Correct Answer: {correct_answer}
Student's Answer: {predicted_answer}

Apply the system rubric: extract final answers, match semantically (order/labels irrelevant), allow small rounding. Use math_verify only for simple symbolic/numeric checks (use 1–1 LaTeX lists). Return **only** the JSON defined above.

# D Evaluated Models and Metadata

This appendix provides a complete list of the LLMs evaluated in our experiments. For each model, we specify:

- **Provider** and exact model name / variant used in evaluation,
- **Model type** (proprietary, open-source, or math/code-specialized),
- Default **prompting style** in our experiments (*Fast* or *Slow*),
- Whether the model natively or indirectly supports our Python sandbox for code execution,
- Tokenizer used for token counting (to normalize across different families),
- Exact **API endpoint or repository commit hash** to ensure reproducibility,
- Path to the prompt template file released in our public repository,
- Any additional notes, such as non-default decoding hyperparameters.

This information ensures that all reported accuracy and token statistics can be faithfully reproduced and verified by independent researchers.

Table 5: Overview of Large Language Models. "Total Parameters" refers to the model's overall size, while "Active Parameters" indicates the parameters engaged during inference for MoE models.

| Model Name | Link | Total Params | Active Params | Architecture | Enable Reasoning | Organization |
|---|---|---|---|---|---|---|
| DeepSeek R1 Guo et al. (2025b) | HF Link | 671B | 37B | MoE + Transformer | Yes | DeepSeek |
| DeepSeek V3 Liu et al. (2024) | HF Link | 671B | 37B | MoE + Transformer | No | DeepSeek |
| DeepSeek V3.1 Liu et al. (2024) | HF Link | 671B | 37B | MoE + Transformer | No | DeepSeek |
| GLM-4.5 Air Zeng et al. (2025) | HF Link | 106B | 12B | MoE + Transformer | Yes | Zhipu AI |
| LLaMA 4 Scout Meta (2025) | HF Link | 109B | 17B | MoE + Transformer | No | Meta |
| Qwen3 8B Yang et al. (2025) | HF Link | 8B | 8B | Dense Transformer | Yes | Alibaba |
| Qwen3 32B Yang et al. (2025) | HF Link | 32B | 32B | Dense Transformer | Yes | Alibaba |
| Qwen3 235B Yang et al. (2025) | HF Link | 235B | 22B | MoE + Transformer | Yes | Alibaba |
| Claude 4 Sonnet Anthropic (2025) | API Link | Undisclosed | Undisclosed | Undisclosed | No | Anthropic |
| Gemini 2.5 Flash Comanici et al. (2025) | API Link | Undisclosed | Undisclosed | Undisclosed | Yes | Google |
| Gemini 2.5 Flash Lite Comanici et al. (2025) | API Link | Undisclosed | Undisclosed | Undisclosed | No | Google |
| Gemini 2.5 Pro Comanici et al. (2025) | API Link | Undisclosed | Undisclosed | Undisclosed | Yes | Google |
| Mathstral 7B Jiang et al. (2023) | HF Link | 7B | 7B | Dense Transformer | No | Mistral AI |
| Qwen2.5-Math 7B Yang et al. (2024) | HF Link | 7B | 7B | Dense Transformer | No | Alibaba |
| Qwen2.5-Math 72B Yang et al. (2024) | HF Link | 72B | 72B | Dense Transformer | No | Alibaba |
| Qwen3-Coder 30B Yang et al. (2025) | HF Link | 30B | 3B | MoE + Transformer | No | Alibaba |

## E    MORE EXPERIMENT DETAILS

To complement the main results, we provide additional figures that examine accuracy and token statistics in greater detail.

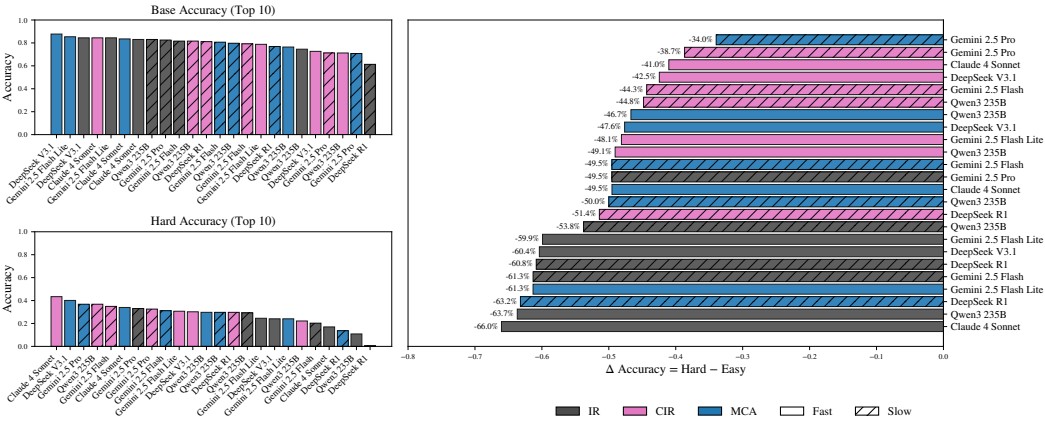

Figure 7: **Accuracy of top-10 models.** Left: Easy and Hard accuracies (ranked separately). Right: $\Delta = $ Hard $-$ Easy. All models exhibit accuracy drops on the Hard set, but the decline is less severe for tool-integrated modes (CIR/MCA).

The first analysis focuses on the top 10 model variants, as shown in Figure 7, ranked by their overall average accuracy across inference modes (IR, CIR, MCA), speeds (Fast/Slow), and both Base and Hard subsets, with all speed and reasoning-type configurations included. Accuracy is computed as the ratio of correct answers to the total, with denominators normalized at the global level, based on the number of questions in the Base and Hard subsets. As expected, all models exhibit negative gaps, reflecting the increased difficulty of the Hard set, though the magnitude of this drop varies across reasoning modes. From the $\Delta$ accuracy plot, it is evident that tool-integrated models show the smallest performance drop, as indicated by the predominance of blue and pink bars in the upper portion of the figure. Moreover, the figure shows that on the Base set most models achieve accuracies above 70%, whereas on the Hard set their performance drops to around 30%.

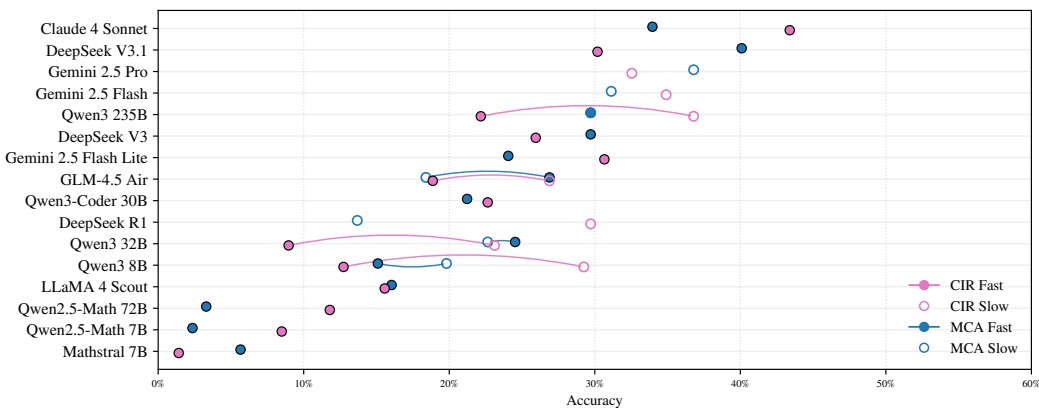

Figure 8: **Dumbbell plots of within-model contrasts on the Hard subset.** Each line connects Fast and Slow variants of the same model across CIR vs. MCA. CIR generally benefits from Slow prompting, while MCA often favors Fast prompting.

As shown in Figure 8, dumbbell plots reveal within-model contrasts on the Hard set by connecting CIR vs. MCA and Fast vs. Slow configurations of the same model. Rows are ordered by mean accuracy. The results indicate a systematic shift: CIR tends to achieve higher accuracy under Slow prompting, consistent with the benefits of more structured reasoning, whereas MCA does not follow

this trend and frequently yields better performance in the Fast setting (e.g., QWEN3-32B, GLM-4.5 AIR).

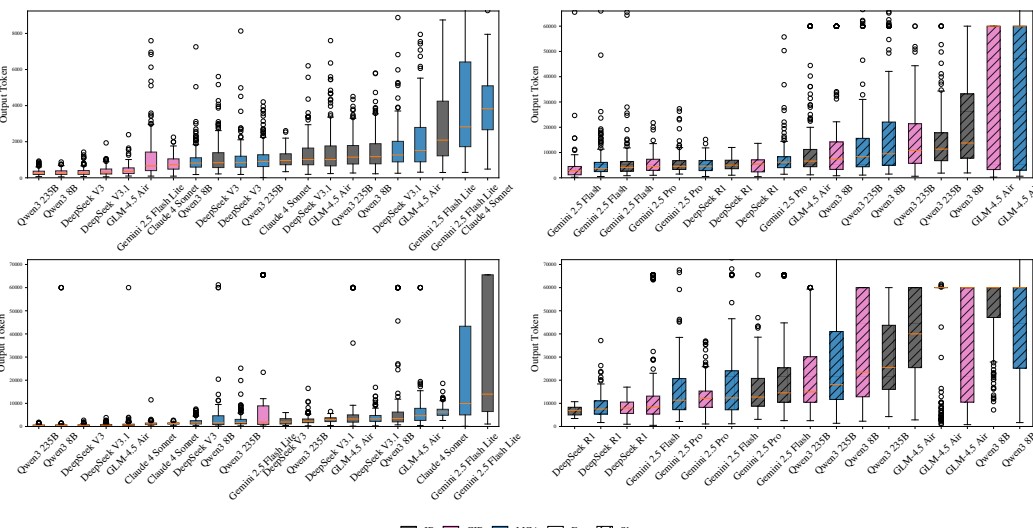

Figure 9: **Boxplots of output token counts by reasoning type and speed.** Shown for Base (top) and Hard (bottom), split into Fast (left) and Slow (right). Slow prompting produces consistently longer outputs, with token-usage rankings stable across difficulty levels.

The distributions of output tokens across different settings are shown in Figure 9, with Base (top) and Hard (bottom) subsets, and Fast (left) versus Slow (right) modes. The coincident upper whiskers in several boxplots arise because multiple generations reached the per-turn output cap of 60k tokens, or in some cases the overall context budget of 128k tokens; these are artifacts of imposed limits rather than genuine features of the distribution. As expected, the right-hand panels (Slow) exhibit higher medians and wider spreads than the left-hand panels (Fast), reflecting greater verbosity under Slow prompting. Comparing the top and bottom rows, the relative ranking by median token count is largely stable—especially in the Fast setting—suggesting that token usage is primarily driven by model style rather than problem difficulty.

## F  BENCHMARK SAMPLES

In this section, we provide representative examples from our benchmark dataset to illustrate the variety and complexity of mathematical problems covered, including both the mathematical formulation and corresponding implementation code.

---

**Sample 1: Nurse Scheduling Optimization (Base Level)**

**Problem:**
A hospital must schedule 4 nurses to cover 3 time blocks: morning, afternoon, and night. Each block requires **[18.5, 20.0, 16.25]** total hours respectively.
Constraints:

- No nurse may work more than **20 hours** in total
- Any assigned shift must be either **0 or at least 3.0 hours** (no short shifts)
- Goal: minimize total scheduled working hours across all nurses

**Mathematical Formulation:**
Since the objective penalizes working hours, the optimal solution meets each demand exactly. Therefore the minimum equals: $18.5 + 20.0 + 16.25 = 54.75$.

**Implementation:**

```python
from scipy.optimize import linprog
import numpy as np

# Problem data
required_hours = [18.5, 20.0, 16.25]
nurse_num, shift_num = 4, 3
max_hours_per_nurse = 20.0

# Since objective minimizes total hours and constraints only
    provide
# lower bounds on coverage, optimal solution meets demands exactly
total_min_hours = sum(required_hours)
print(f"Minimum total working hours: {total_min_hours}")
```

**Final Answer: 54.75**

---

**Sample 2: Meal Planning with Piecewise Pricing (Base Level)**

**Problem:**
Design a daily meal plan using 8 ingredients with nutrition requirements, integer constraints, and piecewise pricing discounts.

**Key Constraints:** protein $\geq$ 65g, fat $\geq$ 40g, carbs $\geq$ 70g, vitamins $\geq$ 100, minerals $\geq$ 80, calories $\geq$ 1200 kcal with $\geq$ 30% from protein+fat, at least 3 different ingredients, I3 and I6 must be integers.

**Mathematical Formulation:**
Use binary variables for discount triggers and ingredient selection. Split I3 and I6 into low/high tiers to model piecewise pricing exactly.

**Implementation:**

```python
import numpy as np
from scipy.optimize import milp, Bounds, LinearConstraint

# Data: cost, protein, fat, carbs, vitamins, minerals per 100g
cost    = np.array([4, 6, 5, 7, 3, 9, 5, 6], dtype=float)
protein = np.array([10,12, 6, 8, 4,14, 8, 5], dtype=float)
fat     = np.array([2, 8,10, 2, 4, 6, 4, 7], dtype=float)
carb    = np.array([8, 2, 4,14,12, 2,10, 7], dtype=float)
vit     = np.array([15, 8,10,16, 8,12, 9, 7], dtype=float)
minr    = np.array([8,12, 6, 8,14,10,11,13], dtype=float)

# Variables: x1,x2,x3_lo,x3_hi,x4,x5,x6_lo,x6_hi,x7,x8,y3,y6,z1-z8
nvar = 20
c = np.zeros(nvar)  # objective coefficients
c[:2] = [4.0, 6.0]   # x1, x2
c[2:4] = [5.0, 4.5]  # x3_lo (normal), x3_hi (discounted)
c[4:6] = [7.0, 3.0]  # x4, x5
c[6:8] = [9.0, 7.2]  # x6_lo (normal), x6_hi (discounted)
c[8:10] = [5.0, 6.0]  # x7, x8

# Bounds and integrality
lb = np.zeros(nvar)
ub = np.full(nvar, 1000.0)
ub[2], ub[6] = 7.0, 5.0  # upper bounds for low tiers
ub[10:] = 1.0  # binary variables

integrality = np.array([0,0,1,1,0,0,1,1,0,0,2,2,2,2,2,2,2,2,2,2])

# Build constraints systematically
A, lb_con, ub_con = [], [], []

# Discount logic for I3: x3_hi >= 8*y3, x3_hi <= 1000*y3, x3_lo <=
    7*(1-y3)
row = np.zeros(nvar); row[3] = 1; row[10] = -8
A.append(row); lb_con.append(0.0); ub_con.append(np.inf)
row = np.zeros(nvar); row[3] = 1; row[10] = -1000
A.append(row); lb_con.append(-np.inf); ub_con.append(0.0)
row = np.zeros(nvar); row[2] = 1; row[10] = 7
A.append(row); lb_con.append(-np.inf); ub_con.append(7.0)

# Similar discount logic for I6
row = np.zeros(nvar); row[7] = 1; row[11] = -6
A.append(row); lb_con.append(0.0); ub_con.append(np.inf)
row = np.zeros(nvar); row[7] = 1; row[11] = -1000
A.append(row); lb_con.append(-np.inf); ub_con.append(0.0)
row = np.zeros(nvar); row[6] = 1; row[11] = 5
A.append(row); lb_con.append(-np.inf); ub_con.append(5.0)

# Selection constraints and nutrition requirements...
```

```
# [Complete constraint matrix construction]

# At least 3 ingredients
row = np.zeros(nvar); row[12:20] = 1.0
A.append(row); lb_con.append(3.0); ub_con.append(np.inf)

# Solve MILP
A = np.vstack(A)
constraints = LinearConstraint(A, np.array(lb_con), np.array(ub_con
    ))
bounds = Bounds(lb, ub)

result = milp(c=c, constraints=constraints, integrality=integrality
    ,
              bounds=bounds)
print(f"Optimal cost: ${result.fun:.2f}")
```

**Final Answer: 39.45**

**Sample 3: Block Tridiagonal System Solution (Hard Level)**

**Problem:**
Let $n = 5000$. Matrix $A$ is block tridiagonal: diagonal blocks $B = \text{tridiag}(-1, 4, -1)$ (10×10), off-diagonal blocks are 10×10 identity matrices. Solve $Ax = b$ where $b = (1, 1, \ldots, 1)^T$ and report $\|x\|_2$.

**Mathematical Approach:**
Use Kronecker sum decomposition: $A = I_p \otimes B + T_p \otimes I_m$ where $p = 500, m = 10$.

**Implementation:**

```python
import numpy as np

# Problem parameters
p, m = 500, 10

# Eigenvalues of tridiagonal matrices
def tridiag_eigenvals(n, a, b, c):
    """Eigenvalues of tridiag(a,b,c) matrix"""
    k = np.arange(1, n+1)
    return b + 2*a*np.cos(np.pi*k/(n+1))

# Eigenvalues
lam_T = tridiag_eigenvals(p, 1, 0, 1)  # T_p eigenvalues
lam_B = tridiag_eigenvals(m, -1, 4, -1)  # B eigenvalues

# Projection coefficients for ones vector
def ones_projection_coeff(n):
    """Coefficients <1, u_k> for sine eigenvectors"""
    k = np.arange(1, n+1)
    theta = np.pi * k / (n + 1)
    S_n = (np.sin(n*theta/2) * np.sin((n+1)*theta/2)
           / np.sin(theta/2))
    return np.sqrt(2/(n+1)) * S_n

alpha = ones_projection_coeff(p)  # for size p
beta = ones_projection_coeff(m)   # for size m

# Compute ||x||_2^2 using spectral decomposition
eigenvals = lam_T[:, None] + lam_B[None, :]  # p x m matrix
coeffs_sq = (alpha[:, None]**2) * (beta[None, :]**2)
norm_sq = np.sum(coeffs_sq / (eigenvals**2))
norm_2 = np.sqrt(norm_sq)

print(f"||x||_2 = {norm_2:.6f}")
```

**Final Answer: 17.055084**

**Sample 4: Nonlinear Integral Equation (Hard Level)**

**Problem:**
Solve for all $x \in (0, 20)$ such that $f(x) = 0.1$ where $f(x) = \int_0^1 \frac{\sin(xt)}{\sqrt{1-t^4}} \, dt$.

**Implementation:**

```python
import numpy as np
from scipy.integrate import quad
from scipy.optimize import brentq

def integrand(t, x):
    """Integrand function with singularity handling"""
    if abs(1 - t**4) < 1e-15:
        return 0  # Handle singularity at t=1
    return np.sin(x * t) / np.sqrt(1 - t**4)

def f(x):
    """Target function f(x) - 0.1"""
    integral, _ = quad(lambda t: integrand(t, x), 0, 1,
                       limit=100, epsabs=1e-12)
    return integral - 0.1

# Scan for sign changes to locate roots
x_scan = np.linspace(1e-6, 20.0, 4000)
y_scan = [f(x) for x in x_scan]

roots = []
for i in range(len(x_scan)-1):
    if y_scan[i] * y_scan[i+1] < 0:
        # Found sign change, refine with Brent's method
        try:
            root = brentq(f, x_scan[i], x_scan[i+1],
                          xtol=1e-12, maxiter=100)
            roots.append(root)
        except:
            pass

# Remove duplicates and format
unique_roots = []
for r in sorted(roots):
    if not unique_roots or abs(r - unique_roots[-1]) > 1e-6:
        unique_roots.append(r)

result = [f"{r:.6f}" for r in unique_roots]
print("Solutions:", ",_".join(result))
```

**Final Answer:** 0.127544, 4.337245, 6.990414, 10.238756, 13.486897, 16.338351, 19.907142

## G    MORE ERRORS DISCUSSION

To further illustrate the failure modes of LLMs, Table 6 contrasts representative incorrect and corrected solutions across five categories: Implementation, Reasoning, Understanding, Syntax, and Format. These examples highlight how errors can occur at different stages of the reasoning-to-code pipeline. Implementation errors typically arise when mathematical expressions are mistranslated into code, while reasoning errors reflect deeper flaws in the formulation of the mathematical abstraction itself. Understanding errors stem from misinterpretations of the problem statement. Syntax errors, by contrast, are low-level violations of programming grammar, and format errors capture failures to adhere to required output structures (e.g., producing invalid JSON).

Table 6: **Error contrast with cases**: Implementation, Reasoning, Understanding, Syntax and Format.

| Error Type | Incorrect | Correct |
|---|---|---|
| **Implementation** (Mistakes when turning math into code) | `f_double_prime = ((6*x*(1 + atan(x**3))*(1 - 3*x**6)) - 9*x**4) / (((1 + x**6)**2) * (1 + atan(x**3))**2)` | `f_double_prime = (6*x*(1 - 2*x**6)*(1 + atan(x**3)) - 9*x**4) / (((1 + x**6)**2) * (1 + atan(x**3))**2)` |
| **Reasoning** (Wrong mathematical model or misuse of definitions) | Chose $x = 4$ as the solution of $x^2 = 16$ 

 *Assumed uniqueness, but both $+4$ and $-4$ map to 16.* | Use the integral branch of $k(x)$ since the inverse is not unique. 

 $\int_0^{16}(t^{1/3} + 2)\,dt = 24\sqrt[3]{2} + 32$ |
| **Understanding** (Misreading the problem statement) | `# Misread the axis:  rotated about y-axis (''y=0'')` 
 $V = \pi \int_0^2 [\phi(f(x))]^2\,dx$ but treated as rotation around y-axis | `# Correct:  rotation is about x-axis` $(y = 0)$ 
 $[\phi(f(x))]^2 = f(x),$ so $V = \pi \int_0^2 f(x)\,dx$ |
| **Syntax** (Pure code syntax mistakes) | `if num_boys` | `if num_boys <= total_boys and num_girls <= total_girls:` |
| **Format** (Violating required output format) | `{ answer:  42,` 
 `"explanation":` 
 `"value is doubled"}` | `{ "answer":  42,` 
 `"explanation":` 
 `"value is doubled" }` |

Beyond these localized cases, we also observe systematic error modes tied to specific models. For CLAUDE, a recurring issue is the *recursion-limit failure*, which appeared in roughly ∼100 problems. This behavior seems linked to the model's optimization for extended context handling: when confronted with difficult integrals, it tends to enter repeated reasoning–execution loops. Eventually, the controller exhausts its recursion depth, terminating the run before a final answer can be produced.

---

**Claude: Recursion Limit Exceeded**

$$y(x) = \frac{1}{5}\Big(\sqrt{x^4 + 2x^2 + 1} + \ln(x^2 + 1)\Big), \quad x \in [0, 3].$$

**Reference value (for verification).**
Correct arc-length value: `3.8585`.

**Model & Mode.**
`claude_sonnet_4_fast` (inference mode: MCA).

**Observed failure (trace excerpt).**
`Recursion limit of 10 reached without hitting a stop condition.`

**Symptom.**
Run terminated without a final numeric answer. The controller hit the orchestration recursion cap (10) due to repeated planning/execution cycles.

---

In contrast, a different failure mode is characteristic of DEEPSEEK-R1. In approximately 60–70 cases, the model terminated prematurely without producing a final answer, even though token and time limits were not exceeded. Unlike Claude's runaway iteration, DeepSeek often halts mid-way

under CIR/MCA-style prompting, generating partial reasoning text but failing to produce a complete code block or final print statement. This suggests instability in closing the reasoning–to–execution loop, reflecting a model-specific weakness in balancing extended deliberation with structured output requirements.

---

### DeepSeek-R1: Truncated Response without Final Output

Survey problem involving magazine readership with overlaps up to triplets. Required outputs: (a) number of people reading at least one magazine, (b) pair with highest overlap, (c) number of people reading exactly two magazines.

**Reference value (for verification).**
(a) 5247    (b) (7, 8), 2050    (c) 30015

**Model & Mode.**
`deepseek_r1_slow` (inference mode: CIR / MCA).

**Observed failure.**
```
No Python code found in response.  The model generated partial
reasoning text but never produced an executable code block or a
final printed answer.
```

**Symptom.**
Run terminated without producing the required `Final Answer:  ...` line. Neither token usage nor time exceeded the configured limits; instead, the model truncated mid-way before yielding executable output.

---

