# OpenReview forum: "MMRC: Measuring Massive-Computational Math Reasoning with Code in LLMs"
_ICLR.cc/2026/Conference — ICLR 2026 Conference Withdrawn Submission_

### Official Review · Reviewer_uqUr · 2025-10-21

**Soundness:** 2
**Presentation:** 2
**Contribution:** 2
**Rating:** 2
**Confidence:** 4

**Summary:**

MMRC is a 500‑problem benchmark drawn from upper‑undergraduate math—calculus, discrete math, linear algebra, linear programming, mathematical physics, and numerical methods. The problems are deliberately computation‑heavy and tied to code execution, so purely text‑only reasoning rarely suffices. The authors evaluate 120 model variants across three setups: a text‑only baseline (IR), a code‑writing mode where the model outputs a full Python program (CIR), and an agentic loop that iteratively calls a Python interpreter (MCA). The results track intuition: models handle the “base” split far better than the code‑dependent “hard” split; integrating code boosts accuracy on the harder problems but can hurt on easier ones; and CIR and MCA fail for different reasons—CIR often on syntax or implementation details, MCA more on planning and understanding. Overall, the benchmark usefully stresses code‑centric mathematical reasoning and makes error modes clearer.

**Strengths:**

- Novel math–code angle: The paper targets math questions that genuinely require writing and running code. The focus on university-level topics (not K–12 or olympiad puzzles) and problems that demand substantial computation fills a gap that most existing benchmarks don’t cover.

- Problem curation and quality: The dataset feels carefully built. Each problem comes from experts, was iteratively revised, and—by the authors’ account—took several hours per item to adapt, integrate code, and cross-check. That level of effort suggests the tasks are nontrivial and well vetted.

- Evaluation breadth: The experiments cover a wide range of recent LLMs (120 configurations) and compare several reasoning setups. Including two code-executing modes (CIR and MCA) alongside a text-only baseline makes it easy to see when code actually moves the needle and how different strategies behave.

- Analysis and takeaways: The error analysis is informative. CIR tends to fail on brittle implementation details—one syntax/implementation mistake can sink the whole run—whereas MCA is more prone to higher-level reasoning and planning errors. These patterns are clearly evidenced and help explain where current systems struggle.

**Weaknesses:**

1) Scope and contribution
I read this as primarily a benchmark paper. The curation and empirical analysis are useful, but there isn’t an algorithmic contribution. Prior work (e.g., MathCoder, DynaMath, Program of Thoughts, OlympiadBench) has already made the case that execution helps on hard math. If you can sharpen what’s uniquely new here—beyond “more computation-heavy”—that would help. For instance, what diagnostics or annotations does MMRC enable that others don’t?

2) Scale and statistical strength
Five hundred problems (200 base, 300 hard) feels small for fine-grained comparisons across domains/models. With only a few items per subdomain, it’s hard to make strong claims. I’d encourage reporting uncertainty (bootstrap CIs), effect sizes, and simple significance tests. Even a brief power/variance analysis per domain would help readers calibrate how confident to be.

3) Data provenance and contamination
Because items are adapted from public sources, leakage is a real possibility. A basic contamination audit—near-dup search (MinHash/SimHash), web checks for verbatim matches, and, if possible, screening against known pretraining corpora—would go a long way. Publishing hashes, source metadata, and license info would increase trust. Otherwise it’s hard to separate reasoning from memorization, especially for code-heavy items.

4) Fairness and reproducibility in evaluation
The three setups (IR, CIR, MCA) use different decoding and iteration budgets, so they’re not directly comparable. It would be fairer to align by a common budget (tokens, wall-clock, or number of executions) and report all where feasible. Efficiency should include latency and CPU/GPU time, not just tokens. For reproducibility, please pin Python/library versions, seeds, and numeric tolerances, and consider a containerized env. Small sensitivity checks (temperature, retry limits, exec caps) would clarify how robust the conclusions are.

5) Verification and scoring
The LLM-first grading with symbolic/numeric fallback needs clearer acceptance criteria. Please spell out formatting normalization, unit handling, and numeric tolerances (absolute/relative) to avoid false negatives/positives. Canonicalization for equivalent expressions or AST-based checks would help. If an LLM grader is involved, a small human-audited slice or agreement stats would boost confidence. Also state what happens when an answer is programmatically correct but fails formatting.

6) Differentiation from existing benchmarks
Emphasizing computation dependence is worthwhile, but the overall shape feels close to MathCoder/MathCheck/HARDMATH. A stronger case would be cross-benchmark generalization (tune on MMRC, test elsewhere, and vice versa) or tasks uniquely enabled here (e.g., multi-step stateful execution with explicit verification hooks). Clear evidence that MMRC captures capabilities others miss would strengthen the pitch.

7) Findings and practical takeaways
Most results match intuition: execution helps on hard problems and can sometimes hurt easy ones. That’s good to document, but not surprising. It’d be helpful to translate findings into concrete guidance—when to trigger execution, how to detect cases where code harms accuracy, or what an adaptive controller might look like. On the hard set (top models ~43%), an error analysis could clarify whether the gap is problem difficulty, verifier brittleness, or model limits—and what to try next.

Overall, the benchmark looks thoughtfully assembled and likely useful to practitioners. Tightening the contamination story, aligning resource budgets, reporting uncertainty and latency, and showing cross-benchmark value would substantially increase the paper’s impact.

**Questions:**

1. Quantifying computational dependency
- I’d find it helpful if you could quantify each task’s computational demands to separate genuine reasoning from incidental code execution—for example: minimal symbolic/numeric complexity, a rough count of arithmetic steps for a non-code solution, or an estimated minimal reasoning-token length. Reporting standard errors or confidence intervals for each subset would also make the comparisons statistically tighter.

2. Training data contamination and deduplication
- Given the use of public sources, could you briefly document your contamination checks? For instance, searches for near-duplicates in major pretraining corpora (The Pile, Common Crawl, CodeContests, or publicly described OpenAI/Anthropic subsets) using n-gram matching or embedding-based similarity. Even a small analysis would increase confidence that MMRC is testing reasoning rather than recall.

3. Impact of MCA budget constraints
- Since MCA currently caps at 10 rounds and ~30s/512MB, how does accuracy change if these budgets are relaxed (e.g., 20 rounds or 60s)? It would also be useful to see per-category curves (e.g., numerical simulation vs. discrete math) to understand where performance saturates under resource limits.

4. Verification and numerical tolerances
- Could you spell out the grading tolerances for symbolic vs. numeric answers (relative/absolute/ULP), and whether they vary by task? For approximate or floating-point outputs (ODEs, PDEs, simulations), a unified policy would help ensure apples-to-apples comparisons across models and improve reproducibility.

5. Evaluation infrastructure and hidden test set
- Do you plan to host a hidden evaluation server or leaderboard with a sequestered split to discourage overfitting? With ~500 items, such infrastructure would help preserve benchmark integrity and enable fair community comparisons.

6. Cross-benchmark validation
- It would strengthen the case for MMRC’s representativeness to show transfer: do models tuned or prompted on MMRC improve (or degrade) on related math–code benchmarks such as MathCoder, DynaMath, or Program-of-Thoughts?

7. Human baseline and difficulty calibration
- If available, please report a human baseline (e.g., graduate students or instructors): accuracy and average time per problem. This would anchor task difficulty and help interpret whether current model performance is pedagogically and computationally realistic.

8. Execution environment reproducibility
- Because these tasks depend on Python and scientific libraries, could you specify library versions, random seeds, and steps for floating-point determinism? Providing a Docker image or environment hash would make replication substantially easier.

---

### Official Review · Reviewer_sn19 · 2025-10-26

**Soundness:** 2
**Presentation:** 2
**Contribution:** 2
**Rating:** 2
**Confidence:** 4

**Summary:**

This paper introduces MMRC, a benchmark comprising 500 curated problems designed to evaluate complex mathematical reasoning tasks that necessitate code execution. The authors employ a two-stage evaluation protocol, utilizing GPT-4o and the SymPy library, to verify the correctness of the final results.

**Strengths:**

The primary contribution is the dataset itself: 500 university-level, manually-verified mathematical reasoning problems that are integrated with code.

**Weaknesses:**

1. The paper claims the necessity of code execution as a key contribution. However, this is already an implicit requirement in many existing difficult mathematical reasoning benchmarks. SOTA models increasingly rely on code interpreters to achieve high performance on these tasks, which arguably lessens the novel contribution of this benchmark.

2. Despite emphasizing the integration of mathematics and code, the evaluation protocol appears to neglect the "code" aspect. The methodology is limited to comparing the model's final answer against the ground truth (using GPT-4o and SymPy). It does not seem to assess the correctness, style, or efficiency of the generated code itself. Consequently, aside from the data format, the evaluation is indistinguishable from other math benchmarks, reinforcing the concern raised in Weakness 1.

3. The empirical results on the "Base set" suggest nearly all evaluated models achieve high scores; for example, Qwen3 8B-slow (80.2%) performs comparably to the significantly larger Gemini 2.5 Pro-slow (82.5%). This minimal performance gap indicates the Base set lacks sufficient discriminative power to clearly distinguish between the capabilities of different models, limiting its utility as a contribution.

4. The paper presents conflicting figures regarding the number of models evaluated, which creates significant ambiguity. This inconsistency is evident in the following lines:

    a. Line 18: "We evaluate 120 model configurations, spanning open- and closed-source models."

    b. Line 304: "We evaluate 20 representative LLMs..."

    c. Line 480: "Our evaluation of nearly 120 open- and closed-source models..."

5. The benchmark's scope is confined to Python, and it does not explore integration with other programming languages.

**Questions:**

N/A

---

### Official Review · Reviewer_zm6E · 2025-10-26

**Soundness:** 2
**Presentation:** 3
**Contribution:** 2
**Rating:** 2
**Confidence:** 3

**Summary:**

This paper introduces MMRC, a new benchmark of 500 expert-curated problems designed to evaluate the ability of LLMs to integrate mathematical reasoning with code execution. The problems are drawn from university-level subjects like calculus, linear algebra, and numerical computation, and are deliberately constructed to have high computational workloads, rendering text-only solutions infeasible. The authors evaluate a wide array of LLMs using two primary paradigms: Code-Invoked Reasoning (CIR), where the model generates a complete Python script in one shot, and Math-Code Agent (MCA), where the model interacts iteratively with a Python interpreter. The core findings demonstrate that code integration is indispensable for solving the complex "Hard" subset of problems, consistently improving accuracy. Furthermore, a detailed error analysis reveals that the two paradigms expose different model weaknesses: CIR is brittle and prone to syntax and implementation errors, whereas MCA, by resolving low-level code issues, uncovers deeper flaws in the model's abstract reasoning and problem understanding.

**Strengths:**

1. By focusing on university-level problems that are computationally intractable without code, it directly addresses a critical gap left by existing benchmarks that either focus on elementary math or on theoretical problems solvable with text-based reasoning. The data curation and verification pipeline, involving both AI-based checks and multi-stage human review.

2. Another major strength is the well-designed evaluation framework. The distinction between the CIR and MCA modes allows for a nuanced analysis, effectively separating the ability to perform one-shot programmatic planning from the ability to engage in iterative, tool-assisted problem-solving. The detailed error analysis provides actionable insights into why models fail, insightfully contrasting the execution fragility of CIR with the reasoning fragility of MCA.

**Weaknesses:**

1. The novelty of the evaluation methods (CIR and MCA) is somewhat overstated. CIR is conceptually very similar to established techniques like Program-of-Thoughts, and MCA is a standard implementation of an agentic loop akin to ReAct.
2. The crucial distinction between the "Base" and "Hard" subsets feels subjective and is not formally defined. The paper claims Hard problems are "effectively unsolvable without code," but the threshold for this classification seems ambiguous and may not have been applied consistently by all human authors. A more quantifiable criterion for this split would strengthen the benchmark's design. In addition, judging from the effect and the author's discussion, the Base part actually has no value.
3. In the Related Work section, the authors mention that, compared to cutting-edge, high-difficulty math datasets like AIME, Omni-Math, and FrontierMath, MMRC "targets the integration of natural language and code execution, addressing tasks that require significant computational power." I disagree that this constitutes a significant contribution of the datasets mentioned in this article. Existing datasets can also be solved using code and mathematical agents, and some work has employed this approach. Just because existing evaluation methods don't include these two evaluation methods doesn't mean we should dismiss their existence.

**Questions:**

Please provide an explanation why the community still needs to use the MMRC evaluation set when there are AIME, FrontierMath, U-MATH and other evaluation sets that are difficult, cannot be directly inferred and solved by LLM, and have covered different sub-fields of mathematics?

---

### Official Review · Reviewer_APbh · 2025-10-31

**Soundness:** 3
**Presentation:** 3
**Contribution:** 2
**Rating:** 4
**Confidence:** 4

**Summary:**

- The paper introduces MMRC (Measuring Massive-Computational Math Reasoning with Code), a new benchmark designed to evaluate large language models’ ability to integrate mathematical reasoning with code execution. This is unlike prior math benchmarks that emphasize symbolic or text-only reasoning (e.g., GSM8K, MATH, or AIME).
- MMRC focuses on university-level problems across domains such as calculus, discrete mathematics, linear algebra, linear programming, numerical computation, mathematical physics, and scientific computing.
- MMRC comprises 500 curated problems divided into two subsets - (1) MMRC-Base, which is analytically tractable but computationally heavy; and (2) MMRC-Hard, which is unsolvable without code execution.
- The dataset was created and verified through a multi-stage human-AI pipeline involving logic and code checks.
- The paper evaluates 120 model configurations spanning open-source, proprietary, and specialized math/code models under three inference modes - (1) Internal Reasoning (IR) - pure text-based reasoning; (2) Code-Invoked Reasoning (CIR) - single-pass code generation and execution; and (3) Math-Code Agent (MCA) - interactive reasoning with a Python interpreter.

Key findings:
- Code execution significantly improves accuracy on computation-heavy problems but can degrade performance on simpler, analytically solvable ones.
- CIR and MCA show complementary strengths: CIR is brittle to syntax and implementation errors, while MCA reduces code errors but exposes deeper reasoning flaws.
- Code integration can also reduce token usage, acting as a “reasoning compressor.”

**Strengths:**

- The paper addresses an under-explored area - how well large language models integrate mathematical reasoning with programmatic computation. Most existing benchmarks (e.g., GSM8K, MATH, AIME, MATH-500) emphasize symbolic or analytical reasoning only, whereas MMRC explicitly targets code-dependent reasoning tasks requiring numerical computation and algorithmic problem solving.
- The three inference modes - IR, CIR and MCA - offer a good comparative lens for understanding how LLMs reason differently when allowed to invoke code versus when reasoning purely in text.
- Diverse coverage across seven university-level domains - calculus, discrete math, linear algebra, optimization, numerical computation, and physics.
- Thorough and systematic evaluation - 120 model configurations, covering both open-source and proprietary models
- The dataset design follows a multi-stage human–AI validation pipeline, including code and logic checks to ensure correctness and diversity. Each problem was manually curated and validated, which enhances benchmark reliability. The observed low performance on the "hard" set further lends credibility to the quality of the benchmark.
- Empirical findings and the error taxonomy are insightful and provide useful interpretability for why different inference paradigms succeed or fail.
- The paper is logically organized with relevant visualizations and transparent methodology.

**Weaknesses:**

- While the paper empirically compares IR and “code-invoked reasoning”, it does not explain these differences through any deeper cognitive or algorithmic lens. The authors could have analyzed these modes through frameworks like compositional reasoning or planning and self-verification.
- The empirical coverage is impressive, but the analysis is not thorough. The paper reports accuracy trends but does not explain why certain models benefit or degrade under code execution.
- Lacks clear inter-annotator agreement metrics or quantitative validation of correctness coverage. This is unclear - "The original problems, though complex, did not fully meet our requirement" - how was this arrived at?
- The benchmark focuses on math-heavy fields but does not test whether its findings generalize to related reasoning domains.
- Could better illustrate what distinguishes MMRC problems from prior datasets by showing example problems.
- Does not discuss SWE-Bench.

**Questions:**

This is unclear - "The original problems, though complex, did not fully meet our requirement" - how was this arrived at?

"Further analysis of CIR and MCA modes, as well as the impact of the fast vs. slow setting" -- at least a high-level analysis should be included in the main paper.

**Details Of Ethics Concerns:**

Dataset curation and validation involves humans but the paper does not discuss ethical considerations around this.

---

### Note · Authors · 2025-11-12

I have read and agree with the venue's withdrawal policy on behalf of myself and my co-authors.